# ReAugment: Learning to Augment Few-Shot Time Series with Model Zoo Guidance

## Abstract

Time series forecasting, particularly in few-shot learning scenarios, is challenging due to the limited availability of high-quality training data. To address this, we present a pilot study on using reinforcement learning (RL) for time series data augmentation. Our method, ReAugment, tackles three critical questions: which parts of the training set should be augmented, how the augmentation should be performed, and what advantages RL brings to the process. Specifically, our approach maintains a forecasting model zoo, and by measuring prediction diversity across the models, we identify samples with higher probabilities for overfitting and use them as the anchor points for augmentation. Leveraging RL, our method adaptively transforms the *overfit-prone* samples into new data that not only enhances training set diversity but also directs the augmented data to target regions where the forecasting models are prone to overfitting. We validate the effectiveness of ReAugment across a wide range of base models, showing its advantages in both standard time series forecasting and few-shot learning tasks.

## 1 Introduction

Time series forecasting is a critical task with diverse applications, but it faces significant challenges due to the limited availability of high-quality training data. This challenge is further amplified in time-evolving domains with inherent non-stationarity and becomes even more pronounced in few-shot learning scenarios, where the scarcity of data affects the performance of forecasting models. While recent methods have focused on developing deep-learning architectures to capture long-term trends, clinical patterns, and multivariate relationships (Wu et al., 2021; Nie et al., 2023; Liu et al., 2024b), in this work, we explore a learning-based data augmentation method that can be seamlessly integrated with existing forecasting models.

Effective data augmentation requires generating high-quality, diverse training samples. However, in practice, existing forecasting models typically rely on fixed-form data augmentation techniques (Wen et al., 2021; Cheung & Yeung, 2020), which lack data-dependent adaptability and may introduce unexpected noise. Previous learning-based augmentation methods typically involve contrastive learning (Demirel & Holz, 2024) or Mixup combinations (Schneider et al., 2024) to generate new data sequences. However, these methods are **NOT task-oriented**, as the data generation process is not guided by the performance of the downstream forecasting models. In contrast, we argue that enabling a *closed-loop augmentation-forecasting process* is essential, which requires aligning the training objectives of augmentation models with the resulting forecasting performance.

To achieve this, we present **ReAugment**, which uses reinforcement learning (RL) to enable closed-loop data augmentation and enhance the generalizability of few-shot forecasting models. We aim to answer three critical questions: *(1) Which parts of the training set should be augmented? (2) How should the augmentation be performed? (3) What advantages does RL offer compared to existing approaches?* Accordingly, ReAugment is designed to dynamically identify a training subset of **overfit-prone** samples that would benefit most from augmentation and automatically searches for optimal augmentation policies tailored to these samples.

Specifically, we first construct a *forecasting model zoo*, consisting of diverse instances of the same network architecture trained via cross-validation, to identify overfit-prone training samples in need of augmentation. An interesting finding is that training forecasting models exclusively with data

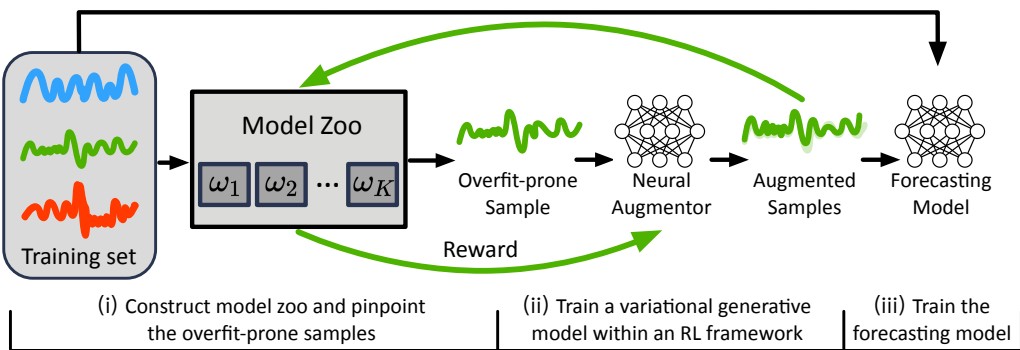

Figure 1: **ReAugment presents an early study on using RL for time series augmentation.**

samples that present **low prediction variance** across the model zoo often leads to better performance. This insight leads us to select the data points with high prediction variance across the model zoo as the root of the subsequent augmentation process.

Based on the identified overfit-prone samples, we leverage a *variational masked autoencoder* (VMAE) to generate augmented training data. The VMAE is trained using an objective function derived from the **backtesting errors of a forecasting model zoo**, where each augmented sample is evaluated across multiple pretrained models. This objective function balances diversity and fidelity: it encourages the generation of samples that lie in regions where models tend to overfit, while maintaining consistency with the original data distribution. Although the objective is differentiable, we adopt the REINFORCE algorithm (Williams, 1992) to train the VMAE, treating its latent space as the action space within an RL framework. This design mitigates the vanishing gradient problem caused by long gradient paths through multiple frozen models in the forecasting model zoo.

In summary, ReAugment forms a closed-loop RL augmentation framework for time series augmentation that couples anchor selection, generative augmentation, and model zoo feedback. It demonstrates substantial improvements across a wide range of base models and datasets, especially in few-shot learning scenarios. Key technical contributions of this work are as follows:

- *Identifying overfit-prone data as augmentation anchors:* We propose a novel method for locating overfit-prone samples by leveraging cross-validation errors from a forecasting model zoo, effectively selecting anchor points with high augmentation potential.
- *Augmentation model design:* We tailor a VMAE for sequential data and treat it as the RL actor to produce sample-aware augmentations.
- *A pilot study of RL-based time series augmentation:* We introduce an RL approach to train the augmentation network, guided by a reward function derived from a forecasting model zoo. This reward encourages the generation of samples in regions where forecasting models are most susceptible to overfitting.

## 2 RELATED WORK

**Transformer-based time series forecasting.** Compared with CNN-based or RNN-based forecasting models (Torres et al., 2021; Wang et al., 2022; Che et al., 2018; Sagheer & Kotb, 2019), recent Transformer-based methods (Li et al., 2019; Wu et al., 2021; Zhou et al., 2021; Liu et al., 2021; Zhou et al., 2022; Zhang & Yan, 2022; Zeng et al., 2023; Cao et al., 2024; Yi et al., 2024) have shown superior performance across a wide range of time series forecasting tasks. For instance, PatchTST (Nie et al., 2023) vectorizes time series data into patches of specified size, which are then encoded through a Transformer, with the model producing forecasts of the desired length via an appropriate prediction head. LSTF-Linear (Zeng et al., 2023) simplifies complex time series forecasting problems and outperforms many Transformers by using a set of remarkably simple one-layer linear models. iTransformer (Liu et al., 2024b) modifies the architecture by adopting components with inverted dimensions, demonstrating superior performance on multivariate time series data. Unlike these methods, we introduce a data augmentation method that can be broadly combined with existing forecasting models.

**Few-shot time series forecasting.** In many real-world applications, obtaining sufficient time series data can be challenging, especially in scenarios where data is scarce, noisy, or difficult to col-

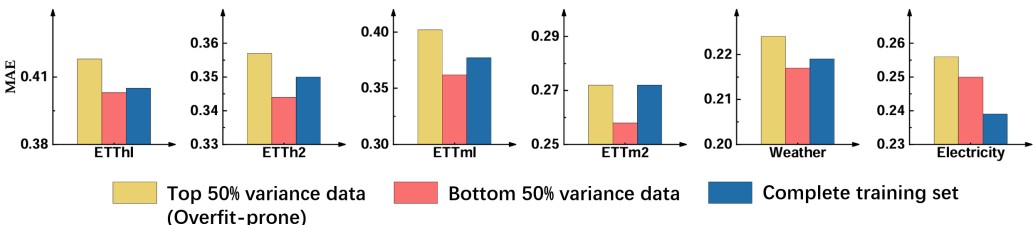

Figure 2: **Preliminary findings on overfit-prone data.** We compare the performance of *iTransformer* trained with different splits of the original training set, which are divided based on the variance of prediction errors across the forecasting model zoo.

lect (Dooley et al., 2023; Xu et al., 2024; Jiang et al., 2023; Yuan et al., 2024). Recent literature has introduced large foundation models specifically designed for time series forecasting (Das et al., 2024; Jin et al., 2024; Bian et al., 2024; Ekambaram et al., 2024; Liu et al., 2024a;c; Pan et al., 2024), often evaluating these models under zero-shot domain generalization settings. However, in preliminary experiments, we found that as the distribution gap between training and testing data increases (*e.g.*, when data comes from different domains), the generalization performance of these models significantly decreases.

**Time series augmentation.** Iglesias *et al.* (Iglesias et al., 2023) have presented a taxonomy of augmentation techniques. Simple augmentation methods involve time, frequency, and magnitude domain transformation techniques such as slicing (Cao et al., 2020), frequency warping (Cui et al., 2015), and jittering (Flores et al., 2021). The second category is the learning-based methods, including those based on contrastive learning (Demirel & Holz, 2024), and Mixup combinations (Schneider et al., 2024). Additionally, advanced generative models have been employed to generate realistic time series data, including the GAN-based (Yoon et al., 2019; Liao et al., 2020), VAE-based (Sohn et al., 2015; Li et al., 2020), and Diffusion-based (Huang et al., 2023) methods. The generated samples can be used for further training of the forecasting models. RL-based sample selection has been explored in other domains (Huang et al., 2024; Yang et al., 2025), but these methods primarily decide which predefined augmentation or sample to use. In contrast, we present a pilot study on applying RL to time series data augmentation, where the agent optimizes a task-aware generative augmentation policy directly within a continuous latent space, enabling fine-grained perturbations beyond discrete operation choices.

## 3 OVERFIT-PRONE DATA AS AUGMENTATION ANCHORS

### 3.1 IDENTIFYING OVERFIT-PRONE DATA WITH A MODEL ZOO

**Motivation.** In time series forecasting, particularly in few-shot learning scenarios, a fact is that forecasting models tend to overfit certain regions of the training data. Intuitively, we aim to generate additional data around these overfit-prone points to improve the model's generalization on nearby data distributions. Therefore, the first challenge we need to address is **identifying overfit-prone data**, *i.e.*, determining which parts of the training set are most likely to benefit from data augmentation. The selected data can then serve as **anchor points** during the augmentation process, acting as input to the generative augmentation model. To achieve this, we measure the cross-validation errors from a batch of forecasting models (termed a *model zoo*) to pinpoint the overfit-prone samples.

**Forecasting model zoo.** To construct the forecasting model zoo, we divide the training set into $k$ parts and perform $k$-fold cross-validation. In this way, we obtain $k$ sets of model parameters for the same network architecture, *e.g.*, iTransformer (Liu et al., 2024b), which we denote as $\mathcal{M}$.

**Model-zoo variance for anchor points selection.** For each data point $x$, we evaluate the *prediction errors* (MSE) of the $k-1$ models that were not trained on the corresponding subset, along with the *training error* of the model trained on that fold. We then compute the variance of these errors across all $k$ models, denoted as $\text{Var}(x; \mathcal{M})$, which captures the inconsistency in predictions made by models trained on different subsets. All data points are subsequently ranked based on this *model-zoo variance*. Intuitively, data points with high model-zoo variance are more likely to lie in overfit-prone regions where models struggle to generalize, making them ideal candidates for data augmentation to improve generalization. Below, we provide empirical evidence to support this claim and assess whether variance can indeed distinguish stable regions from unstable, overfit-prone ones.

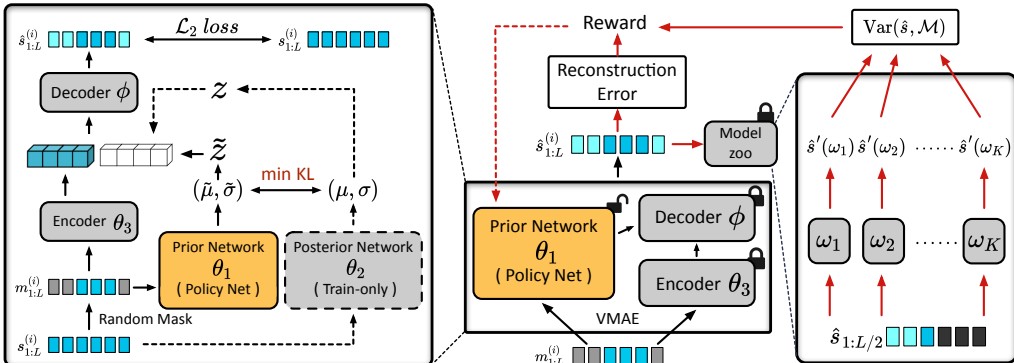

Figure 3: **Architecture of ReAugment.** **Left:** ReAugment pretrains a VMAE as the augmentation backbone, modeling the original distribution of overfit-prone data. **Right:** An RL framework finetunes the VMAE prior network, using its latent space as the action space, guided by a reward function that promotes diverse sample generation around overfit-prone regions.

### 3.2 EMPIRICAL FINDINGS ON OVERFIT-PRONE DATA

**Preliminary experiments.** To investigate the impact of the overfit-prone samples on the training process, we conduct the following experiments: We train forecasting models[1] separately using only Group A (the top $50\%$ subset with higher model-zoo variance) and Group B (the bottom $50\%$ subset with lower model-zoo variance). We evaluate the models on the same test set. The experiments are conducted on classic real-world multivariate benchmarks, including *ETT*, *Weather*, and *Electricity*. The forecasting results are shown in Figure 2, where a lower MAE indicates more accurate predictions. Our results demonstrate that the choice of training data has a significant impact on final performance: across all benchmarks, models trained on Group B (lower variance) consistently outperformed those trained on Group A (higher variance), with substantial margins.

**Insights.** These findings suggest that model-zoo variance can effectively identify overfit-prone samples, and an increased proportion of such samples negatively affects the training quality of forecasting models. The subset analysis not only confirms the validity of the variance-based criterion but also provides a principled explanation for why focusing augmentation on overfit-prone samples yields greater improvements than augmenting the entire dataset Motivated by this insight, our method splits the training set into two subsets based on model-zoo variance and uses the top $50\%$ subset as anchor points for data augmentation. New data is generated around the distributions of these overfit-prone samples, allowing the forecasting model to learn more generalizable patterns.

## 4 REAUGMENT

### 4.1 OVERALL TRAINING PIPELINE

Based on preliminary findings regarding overfit-prone data, we recognize the importance of identifying such samples and using them as anchor points for data augmentation. This approach can help prevent the model from overfitting to specific patterns. By doing so, our method effectively tackles the challenges of data scarcity, leading to improved generalization performance in few-shot learning scenarios. The training pipeline of our approach consists of three stages:

- Stage A: Train a probabilistic generative model to initialize the neural augmentor using overfit-prone samples. Implemented as a VMAE, the model takes partially masked time series data and corresponding absolute timestamps in the dataset as inputs to reconstruct the complete data.
- Stage B: Finetune the VMAE using an RL algorithm, enabling it to generate augmented data that goes beyond merely replicating the original data distribution.
- Stage C: Train the forecasting model using both the original and the augmented data.

Specifically, we augment the top $50\%$ overfit-prone samples, tripling the size of the original training set. This pipeline ensures that data augmentation targets overfit-prone samples, addressing their weaknesses and improving forecasting model performance in data-scarce scenarios.

---

[1]We select iTransformer as the preferred model for these experiments.

## 4.2 Variational Masked Autoencoder as the RL Actor

By using a probabilistic generative model as the neural augmentor, denoted as $\hat{s}_{1:L}^{(i)} \sim G(s_{1:L}^{(i)}, z^{(i)})$, we can generate an infinite amount of data by learning a transformation function based on the overfit-prone data $s_{1:L}^{(i)}$, where $i$ is the data index and $L$ is the length of the data sequence. The key is to learn an appropriate distribution of the latent variable $z$, balancing the diversity of the augmented data with its similarity to the original data. For simplicity, we omit the data index in the following notations. The initial learning step involves optimizing $G$ with a data reconstruction objective, minimizing the divergence between the masked data and the original data distribution.

We design a Variational Masked Autoencoder (VMAE), as illustrated in Figure 3(left), which takes masked time series data $m_{1:L}$ with absolute timestamp $t_{1:L}$ in the whole dataset as input and outputs the complete corresponding data. The entire architecture contains four modules, including (i) the prior network, which learns the prior distribution of $z$ based on the masked data, (ii) the posterior module, which learns the posterior distribution of $z$ based on the original data, (iii) the data encoder, which extracts significant features from $m_{1:L}$ with $t_{1:L}$, (iv) the decoder, which generates data from the encoding features and the latent variables. These modules are parametrized by $\theta_{1:3}$ and $\phi$:

$$\text{Prior:} \quad \tilde{z} \sim p(m_{1:L}, t_{1:L}; \theta_1), \quad \text{Posterior:} \quad z \sim q(s_{1:L}, t_{1:L}; \theta_2),$$
$$\text{Encoder:} \quad u = \text{Enc}(m_{1:L}, t_{1:L}; \theta_3), \quad \text{Decoder:} \quad \hat{s}_{1:L} = \text{Dec}\left(\text{concat}\left(u, z\right); \phi\right). \tag{1}$$

We draw the latent variables, which control the diversity of the generated data, from parametrized Gaussian distributions. The mean and standard deviation of these distributions are produced by the prior and posterior modules. We minimize the Kullback–Leibler (KL) divergence between the prior and posterior distributions. After this training stage, we replace the posterior $z$ with the prior $\tilde{z}$ as input to the decoder. The overall objective function is

$$\mathcal{L} = \mathbb{E}_{s \sim D_s} \|\hat{s}_{1:L} - s_{1:L}\|_2^2 + \beta\, \mathcal{D}_{KL}\left(q(z \mid s_{1:L}, t_{1:L}) \,\|\, p(\tilde{z} \mid m_{1:L}, t_{1:L})\right), \tag{2}$$

where $D_s$ is the set of the overfit-prone data obtained by measuring the prediction diversity over the model zoo. In line with previous work, we use the $\mathcal{L}_2$ loss to measure the reconstruction error. Notably, the posterior module is used exclusively to constrain the prior learner and is not utilized in subsequent training stages.

A key aspect of our method is how we handle the absolute timestamps $t_{1:L}$ during the training and data augmentation phases, respectively. While VMAE is initially trained using the original absolute timestamps from the data, during the data augmentation phase in Stages B&C, we modify these timestamps by sampling them from the test set's time range, rather than from the training time range. Empirically, this technique allows us to generate augmented samples that are more closely aligned with the distribution of the test set.

In general, the VMAE design can be integrated into any encoder-decoder-based time series forecasting architecture. In this work, we specifically adopt the encoder and decoder (implemented as a linear projector) from iTransformer (Liu et al., 2024b) for feature extraction and decoding.

## 4.3 REINFORCE Guided by Model Zoo Predictions

Few-shot time series forecasting is particularly prone to distribution imbalance and shifts, leading to sparse regions that increase overfitting risk. Thus, merely replicating the original data distribution in Stage B is insufficient. To address this, we propose an RL-based framework to expand the distribution around overfit-prone samples, aiming to better cover test patterns and avoid trivial solutions. The augmented data should (i) *increase diversity near overfit-prone regions*, and (ii) *remain close to the original distribution to avoid noise*.

**RL formulation.** The RL component in ReAugment is formulated as a one-step contextual bandit, rather than a multi-step Markov decision process (MDP). Each episode consists of conditioning on a masked time series sample with associated timestamps $(m_{1:L}, t_{1:L})$ as the *state*, sampling a latent code $z$ from the VMAE prior network as the *action*, and decoding it to produce an augmented sample $\hat{s}_{1:L}$. The episode terminates immediately, and a scalar reward is computed based on prediction variance and fidelity. As no temporal credit assignment is required, we set the horizon $T = 1$ and discount factor $\gamma = 0$. This formulation is sufficient for the augmentation task while maintaining stable and efficient policy optimization.

**Reward design.** Intuitively, the variance of prediction errors across the model zoo reflects the degree of overfitting at a given data point. The model zoo $\mathcal{M}$ consists of $K$ network instances of the same network architecture with pretrained parameters $\omega_{1:K}$, constructed via $K$-fold cross-validation. Each data sample is used as test data for one model and as training data for the other $K - 1$ models. We estimate overfitting at each training sequence by computing the variance of prediction errors across the $K - 1$ models that were trained on it. Similarly, for the augmented data $\hat{s}_{1:L}$, we compute the *model-zoo variance* $\mathrm{Var}(\hat{s}_{1:L}, \mathcal{M})$ by

$$\mathrm{Var}(\hat{s}_{1:L}, \mathcal{M}) = \frac{1}{K-1} \sum_{k=1}^{K-1} \left( \|\hat{y}_k(\hat{s}_{1:L}) - y(\hat{s}_{1:L})\|_2^2 - \bar{e} \right)^2,$$

$$\text{s.t.} \quad \bar{e} = \frac{1}{K-1} \sum_{k=1}^{K-1} \|\hat{y}_k(\hat{s}_{1:L}) - y(\hat{s}_{1:L})\|_2^2, \tag{3}$$

where $\hat{y}_k(\hat{s}_{1:L})$ represents the prediction result from the $k$-th forecasting model.

However, a practical challenge in differentiable augmentation learning is that the gradient must be propagated through multiple forecasting models in the model zoo. This results in unstable training and vanishing gradient issues, caused by the aggregation or possible conflicts of gradient signals along multiple paths. To address this, we finetune the neural augmentor using the REINFORCE algorithm, treating the latent space from the VMAE prior module as the action space. Specifically, we optimize the prior module as a policy network while keeping other components fixed, aiming to maximize the following reward function:

$$r = \frac{1}{1 + e^{-\eta \cdot f(\hat{s}_{1:L})}}, \quad \text{s.t.} \ f(\hat{s}_{1:L}) = \frac{\mathrm{Var}\left(\hat{s}_{1:L}, \mathcal{M}\right)}{\|\hat{s}_{1:L} - s_{1:L}\|_2^2}. \tag{4}$$

Based on this reward function, the policy network (*i.e.*, the prior network in VMAE) is optimized as follows via gradient ascent, where $\alpha$ is the learning rate: $\theta_1 \leftarrow \theta_1 + \alpha \cdot r \cdot \nabla_{\theta_1} \log p\left(\tilde{z} \mid m_{1:L}, t_{1:L}; \theta_1\right)$. The reward function encourages augmented samples to lead to increased predictive variance across the model zoo, while simultaneously constraining their deviation from the original data. To prevent rewards from saturating near $0$ or $1$, a scaled sigmoid function controlled by the hyperparameter $\eta$ is applied.

## 5 Experiments

### 5.1 Experimental Setups

Following previous work (Wu et al., 2021; Liu et al., 2024b; Nie et al., 2023; Zeng et al., 2023), we evaluate the proposed ReAugment on five publicly available real-world datasets: *ETT* (including 4 subsets), *Traffic*, *Electricity*, *Weather*, and *Exchange*. We employ a fixed lookback length of 96 time steps across all datasets and report the multivariate sequence prediction results with prediction lengths of 96 time steps. We conduct evaluations under two settings: *few-shot* and *standard*, where the latter allows full access to the original training data of the above datasets. Further details on the datasets, model implementation, and hyperparameter analyses are provided in Appendices B–C.

- *Few-shot setup*: we simulate scenarios with limited training data to assess the model's ability to handle data scarcity. We reduce the training set size to either $10\%$ or $20\%$ of the full dataset, depending on the dataset characteristics. The few-shot training data corresponds to the earliest portion of the time series, ensuring a significant distribution shift from the test set. The validation and test sets remain consistent with those used in prior studies.
- *Standard setup*: We follow the configuration from previous work, allowing access to the full training set while keeping the validation and test sets unchanged.

We primarily use iTransformer (Liu et al., 2024b) for the forecasting model, as it has demonstrated strong performance in standard time series forecasting tasks. We also conduct experiments with PatchTST (Nie et al., 2023) and DLinear (Zeng et al., 2023). Unless otherwise specified, we use a model zoo consisting of $4$ cross-validation models.

We compare ReAugment with the following augmentation methods:

- *Gaussian augmentation*: Inspired by prior literature (Iglesias et al., 2023), we use traditional data augmentation methods that apply simple transformations, such as adding Gaussian noise to the raw data. This approach enhances the diversity of the training data by adjusting the controllable variances and means of the added Gaussian noise.

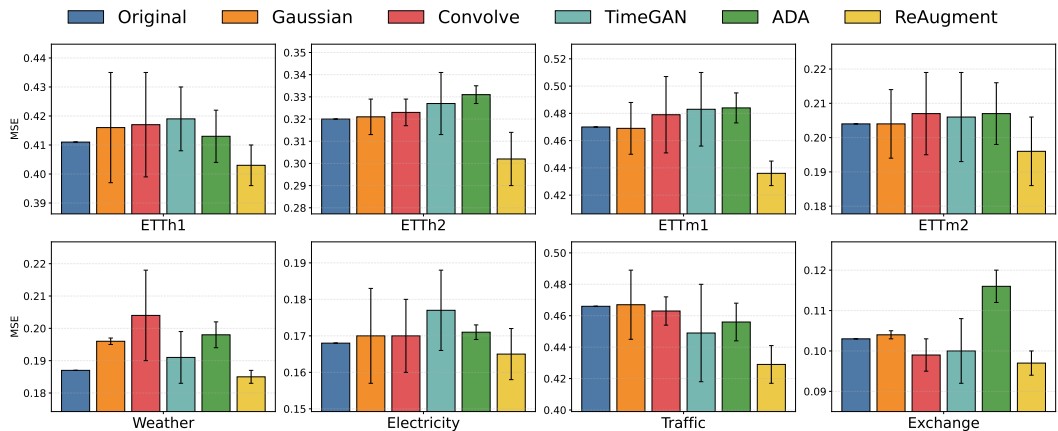

Figure 4: **Few-shot forecasting results in MSE from different augmentation methods.** We consistently use *iTransformer* as the forecasting model and employ the same training seeds. We evaluate each approach over three random seeds and report the mean and standard deviation of the results.

Table 1: **Impact of ReAugment on different forecasting models for few-shot learning.** Each approach is evaluated over three random seeds. Due to page limit, only the mean results are reported here, while both the mean and standard deviation are provided in Table 11 in Appendix E.

| Model | ETTh1 | | ETTh2 | | ETTm1 | | ETTm2 | | Weather | | Electricity | | Traffic | | Exchange | |
|---|---|---|---|---|---|---|---|---|---|---|---|---|---|---|---|---|
| | MAE | MSE | MAE | MSE | MAE | MSE | MAE | MSE | MAE | MSE | MAE | MSE | MAE | MSE | MAE | MSE |
| PatchTST | 0.458 | 0.446 | 0.367 | 0.321 | 0.428 | 0.457 | 0.276 | 0.199 | 0.232 | 0.189 | 0.295 | 0.200 | 0.327 | 0.541 | 0.226 | 0.103 |
| + ReAugment | 0.440 | 0.429 | 0.349 | 0.306 | 0.403 | 0.433 | 0.268 | 0.193 | 0.227 | 0.186 | 0.275 | 0.187 | 0.314 | 0.521 | 0.223 | 0.098 |
| DLinear | 0.435 | 0.408 | 0.402 | 0.356 | 0.442 | 0.471 | 0.303 | 0.219 | 0.277 | 0.212 | 0.307 | 0.215 | 0.452 | 0.724 | 0.226 | 0.104 |
| + ReAugment | 0.422 | 0.388 | 0.369 | 0.334 | 0.431 | 0.462 | 0.297 | 0.216 | 0.276 | 0.212 | 0.305 | 0.216 | 0.406 | 0.663 | 0.224 | 0.099 |

- *Convolve*: We employ another traditional data augmentation method based on the Convolve function in the *Tsaug* library (Wen & Keyes, 2019).
- *TimeGAN* (Yoon et al., 2019): TimeGAN data augmentor combines supervised and adversarial objective optimization. Specifically, through a learned embedding space, the network is guided to adhere to the dynamics of the training data during sampling.
- *ADA* (Schneider et al., 2024): The Anchor Data Augmentation (ADA) method improves domain-agnostic Mixup techniques by generating multiple replicas of modified samples through Anchor Regression, which are then used to create additional training data.

## 5.2 Results of Few-Shot Time Series Forecasting

Under the few-shot setup, we evaluate the effectiveness of ReAugment on different forecasting models, including iTransformer (Liu et al., 2024b), PatchTST (Nie et al., 2023), and DLinear (Zeng et al., 2023). Figure 4 compares the performance of various augmentation methods across several public datasets, using iTransformer as the forecasting model and training with data augmented to three times the original set size. ReAugment achieves the best overall performance, significantly outperforming prior methods. As further shown in Table 1, our method consistently improves prediction accuracy across different forecasting models, indicating that it serves as a flexible design paradigm adaptable to different model backbones.

**New metrics for assessing augmentation benefits.** Evaluating the improved forecasting accuracy achieved through data augmentation provides a direct measure of the effectiveness of our method. However, it does not account for the fact that the impact of data scarcity can vary significantly across different datasets. We propose a new metric that calculates the ratio of the performance promotion achieved through data augmentation in the few-shot setup, compared to the improvement obtained using the full training set. For example, when using MSE, it can be formulated as

$$\mathcal{F}_{\text{MSE}} = \frac{1 - \text{MSE}_{\text{augment}} / \text{MSE}_{\text{few-shot}}}{1 - \text{MSE}_{\text{standard}} / \text{MSE}_{\text{few-shot}}}. \tag{5}$$

As shown in Table 2, a larger value indicates a greater performance improvement due to data augmentation. To make the proposed metrics more interpretable, we emphasize that they measure the

Table 2: **Few-shot time series forecasting results in our new metrics.** These metrics evaluate the effectiveness of data augmentation in the few-shot setting by measuring relative performance improvements, with reference to training on the full dataset. Notably, our approach achieves consistent gains over other augmentation methods. To assess statistical robustness of these new metrics, we additionally report their mean and standard deviation over three runs in Table 13 in Appendix E.

| Dataset | Gaussian | | Convolve | | TimeGAN | | ADA | | ReAugment | |
|---|---|---|---|---|---|---|---|---|---|---|
| | $\mathcal{F}_{\mathbf{MAE}}$ | $\mathcal{F}_{\mathbf{MSE}}$ | $\mathcal{F}_{\mathbf{MAE}}$ | $\mathcal{F}_{\mathbf{MSE}}$ | $\mathcal{F}_{\mathbf{MAE}}$ | $\mathcal{F}_{\mathbf{MSE}}$ | $\mathcal{F}_{\mathbf{MAE}}$ | $\mathcal{F}_{\mathbf{MSE}}$ | $\mathcal{F}_{\mathbf{MAE}}$ | $\mathcal{F}_{\mathbf{MSE}}$ |
| ETTh1 | -10.3% | -20.8% | -24.1% | -25% | -34.5% | -33.3% | -3.4% | -8.3% | **41.4%** | **33.3%** |
| ETTh2 | -25% | -5.3% | -16.7% | -15.8% | -33.3% | -36.8% | -50.0% | -57.9% | **191.7%** | **94.7%** |
| ETTm1 | 3.2% | 0.8% | 22.2% | -7.0% | 15.9% | -10.1% | 17.5% | -10.8% | **47.6%** | **26.4%** |
| ETTm2 | -10.0% | 0.0% | -40.0% | -16.7% | -30.0% | -11.1% | -20.0% | -16.7% | **70.0%** | **44.4%** |
| Weather | -75.0% | -100.0% | -183.3% | -188.9% | -66.7% | -44.4% | -125% | -122.2% | **16.7%** | **22.2%** |
| Electricity | -26.3% | -10.0% | -21.1% | -10.0% | -47.4% | -45.0% | -36.8% | -15.0% | **21.1%** | **15.0%** |
| Traffic | -2.0% | -1.4% | -4.1% | 4.1% | 6.1% | 23.0% | 0.0% | 13.5% | **51.5%** | **50.0%** |
| Exchange | -4.5% | -5.9% | 9.1% | 23.5% | 9.1% | 17.6% | -31.8% | -76.5% | **18.2%** | **35.3%** |

Table 3: **The performance under standard setup with full training set.** Like previous experiments, we use iTransformer as the forecasting model and employ the same random seeds. Each approach is evaluated over three random seeds. Due to page limit, only the mean results are reported here, while both the mean and standard deviation are provided in Table 14 in Appendix E.

| Dataset | Original | | Gaussian | | Convolve | | TimeGAN | | ADA | | ReAugment | |
|---|---|---|---|---|---|---|---|---|---|---|---|---|
| | MAE | MSE | MAE | MSE | MAE | MSE | MAE | MSE | MAE | MSE | MAE | MSE |
| ETTh1 | 0.405 | 0.387 | 0.407 | 0.392 | 0.416 | 0.399 | 0.409 | 0.390 | 0.407 | 0.391 | **0.396**±0.01 | **0.381**±0.01 |
| ETTh2 | 0.350 | 0.301 | 0.352 | 0.307 | 0.356 | 0.303 | 0.348 | 0.299 | 0.347 | 0.297 | **0.346**±0.01 | **0.294**±0.01 |
| ETTm1 | 0.377 | 0.341 | 0.374 | 0.340 | 0.387 | 0.352 | 0.392 | 0.357 | 0.372 | 0.336 | **0.364**±0.01 | **0.328**±0.01 |
| ETTm2 | 0.272 | 0.186 | 0.272 | 0.187 | 0.275 | 0.188 | 0.279 | 0.190 | 0.273 | 0.188 | **0.263**±0.01 | **0.179**±0.00 |
| Weather | 0.219 | 0.178 | 0.227 | 0.187 | 0.265 | 0.210 | 0.219 | 0.177 | 0.222 | 0.180 | **0.206**±0.00 | **0.170**±0.00 |
| Electricity | 0.239 | 0.148 | 0.243 | 0.150 | 0.264 | 0.170 | 0.276 | 0.183 | 0.241 | 0.149 | **0.236**±0.01 | **0.147**±0.01 |
| Traffic | 0.269 | 0.392 | 0.269 | 0.394 | 0.283 | 0.407 | 0.296 | 0.412 | 0.268 | 0.391 | **0.264**±0.01 | **0.388**±0.01 |
| Exchange | 0.206 | 0.086 | 0.208 | 0.087 | 0.210 | 0.087 | 0.210 | 0.087 | 0.209 | 0.087 | **0.204**±0.00 | **0.085**±0.00 |

fraction of the recovered performance gap closed by a given augmentation method relative to a model trained on the full dataset:

- A metric value of $100\%$ indicates that the augmented few-shot model has effectively matched the error of the full-data model; A value greater than $100\%$ suggests that the augmented few-shot model even surpasses the full-data model.

- A value around $0\%$ indicates that the augmentation does not significantly improve upon the original few-shot baseline, while negative values indicate that the augmentation is harmful, as it increases the error compared to training on the original few-shot data alone.

## 5.3 RESULTS WITH FULL ACCESS TO TRAINING SET

ReAugment can also be applied to standard time series forecasting scenarios, where we have full access to the entire training sets. As shown in Table 3, ReAugment delivers significant performance improvements across multiple datasets in the standard setup, highlighting its strong generalizability beyond the few-shot learning context. Other experimental details, such as the lookback and prediction lengths, are consistent with those in the few-shot setup.

## 5.4 COMPARISON WITH FOUNDATION MODELS

The foundation model is an approach to handling few-shot time series prediction tasks, which enhances generalization through a larger number of model parameters and training data. In contrast, ReAugment offers an alternative approach by focusing on the data. Table 4 presents a comparison between ReAugment and the foundation model approach, TimesFM (Das et al., 2024), across multiple datasets. We follow the few-shot setup, with the training data being utilized for supervised finetuning of TimesFM. The results demonstrate that ReAugment (with iTransformer forecasting model) outperforms TimesFM on most dataset, highlighting the effectiveness of a data-driven augmentation approach in improving performance for few-shot time series forecasting.

Table 4: Comparison of ReAugment (iTransformer forecaster) and TimesFM. Electricity, Weather, and Traffic results for TimesFM are not reported as they were used by TimesFM for pretraining.

| Model | ETTh1 | | ETTh2 | | ETTm1 | | ETTm2 | | Exchange | |
|---|---|---|---|---|---|---|---|---|---|---|
| | MAE | MSE | MAE | MSE | MAE | MSE | MAE | MSE | MAE | MSE |
| ReAugment | **0.422** | **0.403** | **0.339** | **0.302** | 0.410 | 0.436 | **0.275** | **0.196** | **0.224** | **0.097** |
| TimesFM | 0.433 | 0.410 | 0.351 | 0.317 | **0.396** | **0.389** | 0.287 | 0.206 | 0.247 | 0.114 |

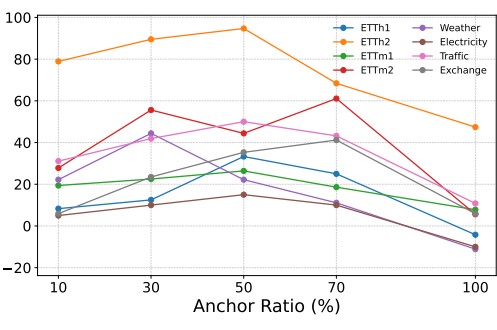 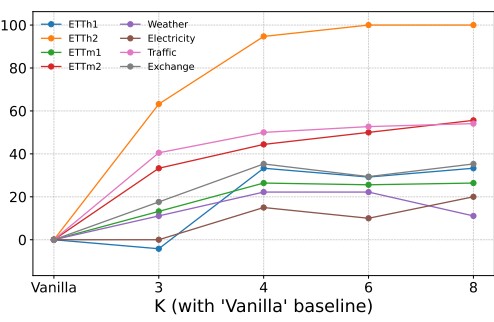

(a) $\mathcal{F}_{\mathbf{MSE}}(\%)$ with different anchor ratio       (b) $\mathcal{F}_{\mathbf{MSE}}(\%)$ with different model zoo size

Figure 5: **Ablation studies.** Left: Effect of the number of overfit-prone samples used as augmentation anchors. Right: Effect of the number of models in the model zoo ($K$) to $\mathcal{F}_{\mathbf{MSE}}$. "Vanilla" denotes the baseline iTransformer trained without augmentation.

## 5.5 MODEL ANALYSES

**Shall we augment all training samples?** Inspired by preliminary findings, we augment the top $50\%$ overfit-prone samples. What would be the impact of augmenting more or fewer samples? To investigate this, we compare augmenting different proportions of overfit-prone samples with augmenting the entire few-shot dataset. For consistency, the total amount of augmented data was kept three times the size of the original training set and applied to the iTransformer model. As shown in Figure 5 (left), directly augmenting the entire few-shot dataset was less effective than data-dependent augmentation, which aligns with our preliminary findings. Furthermore, augmenting different percentages of high-variance samples resulted in varying degrees of improvement, demonstrating that our data augmentation model can adaptively enhance overfit-prone samples, leading to better performance of the forecasting model.

**Impact of model zoo size on performance.** As shown in Figure 5 (right), increasing the number of models in the model zoo generally improves iTransformer's prediction accuracy. This indicates that a larger and more diverse set of models provides richer augmentation signals, leading to more effective few-shot learning. However, the performance gains saturate beyond a certain number of models, suggesting diminishing returns from simply adding more models. These results highlight the importance of carefully selecting a manageable number of models for the augmentation process.

**Impact of RL-based augmentation.** The proposed RL framework enables our model to optimize the latent variable $z$ based on the backtest results across the model zoo, thereby generating augmented data that balances data diversity and similarity to the original data. To evaluate the necessity of RL, we compare our REINFORCE approach with two alternatives during Stage B: (i) removing Stage B entirely, and (ii) directly optimizing the augmentation model via backpropagation using the same objective as the REINFORCE reward. As shown in Figure 6, incorporating RL consistently improves performance in most cases, highlighting its significance for the augmentation quality.

**Why is only the prior network used as the policy network?** We choose the prior network as the policy network because it provides a compact and structured latent action space, which facilitates stable and efficient RL optimization. In contrast, directly using the decoder or the full VMAE involves a high-dimensional action space, leading to slower convergence and lower performance. Empirical results in Table 5 show that the prior-based policy consistently outperforms the other variants across all datasets, validating this design choice.

**Computational costs.** Table 6 reports the training time for VMAE, REINFORCE, and the forecasting model in the few-shot setting. The added time from Stages A and B is acceptable, given

Figure 6: **Impact of RL in Stage B for few-shot forecasting measured by MSE.** For both variants, the forecasting iTransformer is trained using augmented data three times of the original training set.

Table 5: **Comparison of policy network choices in the RL stage.** All settings follow the main few-shot setup. "*Decoder*": finetuning the VMAE decoder during RL training. "*Full VMAE*": finetuning all VMAE parameters. "*Prior*": our final method, which operates in a more compact action space defined on the latent variables, showing consistent performance improvements.

| Policy Net | ETTh1 | | ETTh2 | | ETTm1 | | ETTm2 | | Weather | | Electricity | | Traffic | |
| | MAE | MSE | MAE | MSE | MAE | MSE | MAE | MSE | MAE | MSE | MAE | MSE | MAE | MSE |
|---|---|---|---|---|---|---|---|---|---|---|---|---|---|---|
| Decoder | 0.429 | 0.407 | 0.350 | 0.315 | 0.421 | 0.444 | 0.277 | 0.198 | 0.231 | 0.187 | **0.254** | 0.165 | 0.295 | 0.432 |
| Full VMAE | 0.424 | 0.404 | 0.341 | 0.304 | 0.416 | 0.441 | 0.276 | **0.196** | 0.230 | 0.186 | 0.255 | 0.166 | **0.293** | 0.430 |
| Prior (ours) | **0.422** | **0.403** | **0.339** | **0.302** | **0.410** | **0.436** | **0.275** | **0.196** | **0.229** | **0.185** | **0.254** | **0.165** | **0.293** | **0.429** |

the performance gains, and remains shorter than training the forecasting models. All experiments are conducted on an NVIDIA RTX 3090 GPU. For completeness, we also include two representative augmentation baselines, *TimeGAN* and *ADA*. Among them, TimeGAN requires training a full sequence generator and therefore incurs a higher cost, whereas ADA is relatively lightweight and closer to ReAugment in computational footprint. This comparison shows that ReAugment achieves competitive efficiency while providing superior augmentation quality.

**Hyperparameter analyses.** We further analyze the VMAE mask rate and provide detailed descriptions of other model hyperparameters. See Appendix C for the full results.

**Forecasting time horizons.** In Table 10 in Appendix D, we report ReAugment's performance under varying prediction horizons (96, 192, 336, 720) in the few-shot setup. All experiments use a fixed input sequence length of 96 time steps and employ iTransformer as the forecasting model.

Table 6: **Training costs.** The total training time required for augmentation in our method (Stages A+B) is notably shorter than the time needed for training the forecasting models. We additionally report the training-time overhead of two representative learning-based augmentation baselines, *TimeGAN* and *ADA*. Note that we compare only the training time of Stages A+B with other augmentation methods, since the forecasting model training time is comparable across all methods.

| Dataset | TimeGAN | ADA | Our Method | | Forecasting Model | |
| | | | Stage A: VMAE | Stage B: RL | iTransformer | PatchTST |
|---|---|---|---|---|---|---|
| ETTh1 | 4min | 1min | 1min | 2min | 2min | 5min |
| Electricity | 1h 12min | 18min | 24min | 31min | 1h 33min | 2h 24min |
| Traffic | 3h 48min | 58min | 1h 22min | 1h 53min | 4h 27min | 6h 50min |

## 6 CONCLUSIONS AND LIMITATIONS

In this paper, we proposed ReAugment, a novel data augmentation method driven by reinforcement learning. Key technical contributions include: (i) Identifying overfit-prone data samples that could significantly benefit from augmentation by assessing their prediction diversity across a forecasting model zoo; (ii) Training a variational generative model within an RL framework to transform these overfit-prone samples into new data points, guided by a reward function derived from the performance of the model zoo, thereby enhancing both the quality and diversity of the augmented data. ReAugment significantly boosts forecasting performance while maintaining minimal computational overhead by leveraging a learnable policy to transform the overfit-prone samples.

One unresolved issue in this study is the reliance on multiple pretrained models. The proposed method introduces additional computational overhead due to the need for training VMAE, applying the REINFORCE algorithm, and performing backtesting across the model zoo. We provide a detailed comparison of the computational costs for each training stage in the experimental section.

ETHICS STATEMENT

All experiments in this paper are conducted using publicly available datasets. No private or personally identifiable information is involved, and the work follows standard research ethics guidelines.

REPRODUCIBILITY STATEMENT

The reproducibility of the reported results is supported through multiple components. The main text and appendix specify the model architecture, training setup, and evaluation protocols. Additional implementation details and hyperparameters are provided in the supplementary material. The complete source code is also submitted as supplementary material to facilitate independent verification of all experiments.

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

APPENDIX

## A    OVERALL TRAINING PIPELINE

We present the overall training pipeline in Algorithm 1.

---

**Algorithm 1** Overall training pipeline

---

1: **Given:**  Time series samples from training set $s_{1:L}^{(i)}$
2: **Key problem:**  Which samples should be augmented and how to augment them?
3: // Stage A: Train the VMAE supervised by original data
4: VMAE can be parameterized as $\theta_1, \theta_2, \theta_3, \phi$, all parameters are optimized during the training phase.
5: // Stage B: Data filtering by model zoo variance
6: Pretrain a model zoo and assess on the training set.
7: The top $50\%$ samples with large variance on the model zoo are found from the training set and formulated as $s_{1:L}$.
8: // REINFORCE with model zoo
9: Fixed parameters $\theta_2, \theta_3, \phi$.
10: **while** not converged **do**
11:     Sample batch of time series samples and mask randomly, formulated as $m_{1:L}$.
12:     Input masked data $m_{1:L}$ to pretrained VMAE, calculate the reward $r$ by the output of VMAE and pretrained model zoo.
13:     Update the policy net parameters:

$$\theta_1 \leftarrow \theta_1 + \alpha \cdot r \cdot \nabla_{\theta_1} \log E_{\theta_1} \left( \tilde{\mathbf{z}} \mid m_{1:L}, t \right)$$

14: **end while**
15: Stage C: Train the forecasting model
16: Generate augmented data by ReAugment
17: Further train the forecasting model (*e.g.*, iTransformer) using augmented data

---

## B    DATASET DETAILS

Here is a detailed description of the five experiment datasets:

1. ETT consists of two hourly-level datasets (ETTh) and two 15-minute-level datasets (ETTm). Each of them contains 7 factors of electricity transformers, including load and oil temperature from July 2016 to July 2018.

2. Traffic is a collection of road occupancy rates measured by 862 sensors on San Francisco Bay area freeways from January 2015 to December 2016.

3. ECL collects hourly electricity consumption of 321 clients from 2012 to 2014.

4. The Weather dataset includes 21 meteorological indicators, such as air temperature and humidity, recorded 10 minutes from the weather station of the Max Planck Biogeochemistry Institute in 2020.

5. The Exchange dataset records the daily exchange rates of 8 different countries ranging from 1990 to 2016.

For the standard setup, we follow the data processing method of iTransformer, dividing the dataset into training, validation, and test sets, with this partitioning aligned in chronological order.

In addition, to simulate a scenario with limited training data, we propose the few-shot setup. Specifically, we reduce the training set size to either $10\%$ or $20\%$ of the full dataset, while keeping the validation and test sets the same as in the standard setup. Notably, the training data used consists of the most distant portion of the time series relative to the test set, to simulate a more challenging time series forecasting task. In Table 7, we provide the number of variables (*i.e.*, the feature dimension at a single time point) in each dataset, the total number of time points, and the number of time points within each set of the train-validation-test partitions for both standard and few-shot setup.

Table 7: **Details of the datasets.** *Features* denotes the number of data variables in each dataset. *Time points* refers to the total number of time points in the dataset. *Partition* indicates the number of time points allocated to each subset in the (train, validation, test) splits.

|  | ETTh1 / ETTh2 | ETTm1 / ETTm2 | Traffic |
|---|---|---|---|
| Features | 7 | 7 | 862 |
| Time points (Standard) | 14307 | 57507 | 17451 |
| Time points (Few-shot) | 8443 | 28793 | 8756 |
| Partition (Standard) | (8545, 2881, 2881) | (34465, 11521, 11521) | (12185, 1757, 3509) |
| Partition (Few-shot) | (2681, 2881, 2881) | (5751, 11521, 11521) | (3490, 1757, 3509) |
|  | Electricity | Weather | Exchange |
| Features | 321 | 21 | 8 |
| Time points (Standard) | 26211 | 52603 | 7207 |
| Time points (Few-shot) | 13136 | 21071 | 3528 |
| Partition (Standard) | (18317, 2633, 5261) | (36792, 5271, 10540) | (5120, 665, 1422) |
| Partition (Few-shot) | (5242, 2633, 5261) | (5260, 5271, 10540) | (1441, 665, 1422) |

## C  SENSITIVITY ANALYSIS

### C.1  MASK RATE OF VMAE

The final model uses a mask rate of 0.3, which is kept for all experiments in the main text. As shown in Table 8, we analyze this hyperparameter in a few-shot learning setup.

Table 8: **Performance under different mask rates across datasets.**

| Mask rate | 0 | | 0.1 | | 0.3 | | 0.5 | | 0.7 | |
|---|---|---|---|---|---|---|---|---|---|---|
|  | MAE | MSE | MAE | MSE | MAE | MSE | MAE | MSE | MAE | MSE |
| ETTh1 | 0.436 | 0.412 | 0.428 | 0.409 | **0.422** | **0.403** | 0.425 | 0.406 | 0.435 | 0.413 |
| ETTh2 | 0.363 | 0.320 | 0.353 | 0.314 | **0.339** | **0.302** | 0.340 | 0.303 | 0.342 | 0.305 |
| ETTm1 | 0.439 | 0.470 | 0.423 | 0.448 | 0.410 | 0.436 | **0.408** | **0.435** | 0.412 | 0.439 |
| ETTm2 | 0.282 | 0.205 | 0.277 | 0.200 | **0.275** | **0.196** | **0.275** | 0.195 | 0.276 | 0.198 |
| Weather | 0.234 | 0.190 | 0.229 | 0.186 | 0.229 | 0.185 | 0.229 | 0.185 | **0.228** | **0.184** |

### C.2  ADDITIONAL HYPERPARAMETERS

In Table 9, we provide the hyperparameter details of VMAE and REINFORCE. For the encoder and decoder, we adopt the identical hyperparameters as those employed in iTransformer.

Table 9: **Hyperparameters of ReAugment.**

| Notation | Hyperparameter | Description |
|---|---|---|
| $\alpha$ | 0.001 | Learning rate of REINFORCE |
| $\beta$ | 0.1 | Weight of KL-divergence in the VMAE loss function |
| $L$ | 96 | Time series periods length |
| $\eta$ | 0.01 | Parameters of scaled sigmoid |
| $N$ | 32 | Batch size for VMAE training |

## D  EXPERIMENTS ON PREDICTION HORIZONS

In the main text, we employ a fixed lookback length of 96 time steps across all datasets and report the prediction results with a fixed horizon of 96 time steps. In this section, we explore the impact of different prediction lengths on the performance of our method. Specifically, we evaluate our model's performance with a prediction horizon of varying lengths, including 96, 192, 336, and 720 time steps.

Table 10: **ReAugment performance under varying prediction horizons** (96, 192, 336, 720) **in the few-shot setup.** All experiments use a fixed input sequence length of 96 time steps.

| Forecast Horizon | 96 | | 192 | | 336 | | 720 | |
|---|---|---|---|---|---|---|---|---|
| | MAE | MSE | MAE | MSE | MAE | MSE | MAE | MSE |
| ETTh1 (Raw) | 0.434 | 0.411 | 0.468 | 0.474 | 0.491 | 0.520 | 0.527 | 0.539 |
| + ReAugment | 0.422 | 0.403 | 0.455 | 0.461 | 0.477 | 0.503 | 0.510 | 0.528 |
| ETTh2 (Raw) | 0.362 | 0.320 | 0.419 | 0.397 | 0.453 | 0.448 | 0.469 | 0.449 |
| + ReAugment | 0.339 | 0.302 | 0.401 | 0.378 | 0.428 | 0.419 | 0.436 | 0.423 |
| ETTm1 (Raw) | 0.440 | 0.470 | 0.471 | 0.513 | 0.480 | 0.585 | 0.533 | 0.638 |
| + ReAugment | 0.410 | 0.436 | 0.435 | 0.477 | 0.446 | 0.549 | 0.499 | 0.591 |
| ETTm2 (Raw) | 0.282 | 0.204 | 0.331 | 0.270 | 0.371 | 0.339 | 0.429 | 0.421 |
| + ReAugment | 0.275 | 0.196 | 0.325 | 0.258 | 0.357 | 0.331 | 0.411 | 0.394 |
| Weather (Raw) | 0.231 | 0.187 | 0.273 | 0.235 | 0.319 | 0.315 | 0.372 | 0.384 |
| + ReAugment | 0.229 | 0.185 | 0.270 | 0.232 | 0.307 | 0.310 | 0.364 | 0.370 |
| Electricity (Raw) | 0.258 | 0.168 | 0.270 | 0.181 | 0.289 | 0.197 | 0.340 | 0.248 |
| + ReAugment | 0.254 | 0.165 | 0.265 | 0.179 | 0.281 | 0.192 | 0.332 | 0.239 |
| Traffic (Raw) | 0.318 | 0.466 | 0.329 | 0.484 | 0.340 | 0.497 | 0.371 | 0.542 |
| + ReAugment | 0.293 | 0.429 | 0.300 | 0.451 | 0.319 | 0.467 | 0.348 | 0.509 |
| Exchange (Raw) | 0.228 | 0.103 | 0.327 | 0.195 | 0.473 | 0.358 | 0.725 | 0.892 |
| + ReAugment | 0.224 | 0.097 | 0.319 | 0.190 | 0.471 | 0.355 | 0.713 | 0.879 |

Table 11: **Impact of ReAugment on different forecasting models in the few-shot learning setup.** For the raw data baseline, no standard deviation is reported since the forecasting model is trained with a fixed random seed. We report both the mean and standard deviation for ReAugment.

| Training Data | iTransformer | | PatchTST | | DLinear | |
|---|---|---|---|---|---|---|
| | MAE | MSE | MAE | MSE | MAE | MSE |
| ETTh1 (Raw) | 0.434 | 0.411 | 0.458 | 0.446 | 0.435 | 0.408 |
| + ReAugment | 0.422±0.01 | 0.403±0.01 | 0.440±0.02 | 0.429±0.02 | 0.422±0.02 | 0.388±0.01 |
| ETTh2 (Raw) | 0.362 | 0.320 | 0.367 | 0.321 | 0.402 | 0.356 |
| + ReAugment | 0.339±0.01 | 0.302±0.01 | 0.349±0.01 | 0.306±0.01 | 0.369±0.01 | 0.334±0.01 |
| ETTm1 (Raw) | 0.440 | 0.470 | 0.428 | 0.457 | 0.442 | 0.471 |
| + ReAugment | 0.410±0.01 | 0.436±0.01 | 0.403±0.02 | 0.433±0.02 | 0.431±0.01 | 0.462±0.02 |
| ETTm2 (Raw) | 0.282 | 0.204 | 0.276 | 0.199 | 0.303 | 0.219 |
| + ReAugment | 0.275±0.02 | 0.196±0.01 | 0.268±0.01 | 0.193±0.01 | 0.297±0.00 | 0.216±0.00 |
| Weather (Raw) | 0.231 | 0.187 | 0.232 | 0.189 | 0.277 | 0.212 |
| + ReAugment | 0.229±0.00 | 0.185±0.00 | 0.227±0.00 | 0.186±0.00 | 0.276±0.01 | 0.212±0.00 |
| Electricity (Raw) | 0.258 | 0.168 | 0.295 | 0.200 | 0.307 | 0.215 |
| + ReAugment | 0.254±0.01 | 0.165±0.01 | 0.275±0.01 | 0.187±0.01 | 0.305±0.01 | 0.216±0.01 |
| Traffic (Raw) | 0.318 | 0.466 | 0.327 | 0.541 | 0.452 | 0.724 |
| + ReAugment | 0.293±0.01 | 0.429±0.01 | 0.314±0.02 | 0.521±0.02 | 0.406±0.01 | 0.663±0.02 |
| Exchange (Raw) | 0.228 | 0.103 | 0.226 | 0.103 | 0.226 | 0.104 |
| + ReAugment | 0.224±0.00 | 0.097±0.00 | 0.223±0.00 | 0.098±0.00 | 0.224±0.00 | 0.099±0.00 |

The experiments aim to provide insights into how the performance of ReAugment scales with longer prediction horizons, and to assess whether the model's ability to generalize across different time scales holds up as the forecasting task becomes more challenging. As shown in Table 10, ReAugment consistently yields performance improvements across different prediction horizons.

# E    FULL RESULTS WITH MULTIPLE TRAINING SEEDS

In the main text, for all stochastic data augmentation methods, we conducted experiments using three different random seeds and reported the mean performance. It is worth noting that in all

Table 12: **Few-shot forecasting results using various augmentation methods.** We use iTransformer as the forecasting model and report results across three random seeds.

| Dataset | Original | | Gaussian | | Convolve | |
|---|---|---|---|---|---|---|
| | MAE | MSE | MAE | MSE | MAE | MSE |
| ETTh1 | 0.434 | 0.411 | 0.437±0.02 | 0.416±0.02 | 0.441±0.03 | 0.417±0.02 |
| ETTh2 | 0.362 | 0.320 | 0.365±0.01 | 0.321±0.01 | 0.364±0.01 | 0.323±0.01 |
| ETTm1 | 0.440 | 0.470 | 0.438±0.03 | 0.469±0.02 | 0.426±0.02 | 0.479±0.03 |
| ETTm2 | 0.282 | 0.204 | 0.283±0.01 | 0.204±0.01 | 0.286±0.01 | 0.207±0.01 |
| Weather | 0.231 | 0.187 | 0.240±0.01 | 0.196±0.00 | 0.253±0.01 | 0.204±0.01 |
| Electricity | 0.258 | 0.168 | 0.263±0.01 | 0.170±0.01 | 0.262±0.01 | 0.170±0.01 |
| Traffic | 0.318 | 0.466 | 0.319±0.01 | 0.467±0.02 | 0.320±0.01 | 0.463±0.01 |
| Exchange | 0.228 | 0.103 | 0.229±0.00 | 0.104±0.00 | 0.226±0.01 | 0.099±0.00 |
| Dataset | TimeGAN | | ADA | | ReAugment | |
| ETTh1 | 0.444±0.02 | 0.419±0.01 | 0.435±0.01 | 0.413±0.01 | **0.422**±0.01 | **0.403**±0.01 |
| ETTh2 | 0.366±0.01 | 0.327±0.01 | 0.368±0.00 | 0.331±0.00 | **0.339**±0.01 | **0.302**±0.01 |
| ETTm1 | 0.430±0.03 | 0.483±0.03 | 0.429±0.01 | 0.484±0.01 | **0.410**±0.01 | **0.436**±0.01 |
| ETTm2 | 0.285±0.02 | 0.206±0.01 | 0.284±0.01 | 0.207±0.01 | **0.275**±0.02 | **0.196**±0.01 |
| Weather | 0.239±0.01 | 0.191±0.01 | 0.246±0.00 | 0.198±0.00 | **0.229**±0.00 | **0.185**±0.00 |
| Electricity | 0.267±0.01 | 0.177±0.01 | 0.265±0.01 | 0.171±0.00 | **0.254**±0.01 | **0.165**±0.01 |
| Traffic | 0.315±0.02 | 0.449±0.03 | 0.318±0.01 | 0.456±0.01 | **0.293**±0.01 | **0.429**±0.01 |
| Exchange | 0.226±0.01 | 0.100±0.01 | 0.235±0.00 | 0.116±0.00 | **0.224**±0.00 | **0.097**±0.00 |

Table 13: **Few-shot time series forecasting results in our new metrics.** To assess the statistical robustness of these new metrics, we report their mean and standard deviation over three runs.

| Dataset | Metric | Gaussian | Convolve | TimeGAN | ADA | ReAugment |
|---|---|---|---|---|---|---|
| ETTh1 | $\mathcal{F}_{MAE}$ | -10.3%±6.8% | -24.1%±9.3% | -34.5%±6.4% | -3.4%±3.1% | **41.4%**±**2.7%** |
| | $\mathcal{F}_{MSE}$ | -20.8%±7.3% | -25.0%±6.9% | -33.3%±4.5% | -8.3%±3.9% | **33.3%**±**3.3%** |
| ETTh2 | $\mathcal{F}_{MAE}$ | -25.0%±8.5% | -16.7%±7.7% | -33.3%±8.2% | -50.0%±3.1% | **191.7%**±**6.9%** |
| | $\mathcal{F}_{MSE}$ | -5.3%±4.3% | -15.8%±3.9% | -36.8%±6.8% | -57.9%±2.4% | **94.7%**±**6.2%** |
| ETTm1 | $\mathcal{F}_{MAE}$ | 3.2%±4.6% | 22.2%±3.1% | 15.9%±4.3% | 17.5%±1.8% | **47.6%**±**1.9%** |
| | $\mathcal{F}_{MSE}$ | 0.8%±1.5% | -7.0%±2.3% | -10.1%±2.1% | -10.8%±0.8% | **26.4%**±**0.7%** |
| ETTm2 | $\mathcal{F}_{MAE}$ | -10.0%±8.9% | -40.0%±13.1% | -30.0%±18.8% | -20.0%±8.9% | **70.0%**±**18.3%** |
| | $\mathcal{F}_{MSE}$ | 0.0%±4.9% | -16.7%±5.7% | -11.1%±6.2% | -16.7%±4.4% | **44.4%**±**5.1%** |
| Weather | $\mathcal{F}_{MAE}$ | -75.0%±10.2% | -183.3%±9.1% | -66.7%±8.3% | -125%±2.9% | **16.7%**±**3.1%** |
| | $\mathcal{F}_{MSE}$ | -100.0%±2.4% | -188.9%±14.2% | -44.4%±8.9% | -122.2%±4.1% | **22.2%**±**2.2%** |
| Electricity | $\mathcal{F}_{MAE}$ | -26.3%±4.7% | -21.1%±6.3% | -47.4%±5.8% | -36.8%±6.1% | **21.1%**±**4.4%** |
| | $\mathcal{F}_{MSE}$ | -10.0%±6.0% | -10.0%±4.4% | -45.0%±4.6% | -15.0%±1.3% | **15.0%**±**4.1%** |
| Traffic | $\mathcal{F}_{MAE}$ | -2.0%±1.8% | -4.1%±2.2% | 6.1%±4.4% | 0.0%±1.4% | **51.5%**±**2.2%** |
| | $\mathcal{F}_{MSE}$ | -1.4%±3.0% | 4.1%±1.2% | 23.0%±4.3% | 13.5%±1.7% | **50.0%**±**2.0%** |
| Exchange | $\mathcal{F}_{MAE}$ | -4.5%±0.9% | 9.1%±4.8% | 9.1%±3.8% | -31.8%±2.5% | **18.2%**±**1.5%** |
| | $\mathcal{F}_{MSE}$ | -5.9%±1.4% | 23.5%±2.2% | 17.6%±5.9% | -76.5%±2.1% | **35.3%**±**1.7%** |

experiments, the source of randomness stems solely from the data augmentation methods, rather than the forecasting models. Accordingly, for stochastic augmentation methods, we applied different random seeds during the augmentation process, while keeping the random seed fixed for the training of the forecasting models. Here, we provide the complete results, including both the mean and standard deviation, for the following experiments:

• The impact of ReAugment on different forecasting models under few-shot learning in Table 11;

• The few-shot forecasting performance using different data augmentation methods in Table 12;

• The few-shot performance of different data augmentation methods evaluated using our new metrics, with mean and standard deviation reported in Table 13.

• The performance of ReAugment under the standard setup with the full training set in Table 14.

Table 14: **Comparison under standard setup with full training set.** We use iTransformer as the forecasting model and report the mean and standard deviation across three random seeds.

| Dataset | Original | | Gaussian | | Convolve | |
|---|---|---|---|---|---|---|
| | MAE | MSE | MAE | MSE | MAE | MSE |
| ETTh1 | 0.405 | 0.387 | 0.407±0.02 | 0.392±0.01 | 0.416±0.03 | 0.399±0.02 |
| ETTh2 | 0.350 | 0.301 | 0.352±0.01 | 0.307±0.01 | 0.356±0.01 | 0.303±0.01 |
| ETTm1 | 0.377 | 0.341 | 0.374±0.02 | 0.340±0.02 | 0.387±0.02 | 0.352±0.02 |
| ETTm2 | 0.272 | 0.186 | 0.272±0.01 | 0.187±0.00 | 0.275±0.01 | 0.188±0.01 |
| Weather | 0.219 | 0.178 | 0.227±0.01 | 0.187±0.01 | 0.265±0.01 | 0.210±0.01 |
| Electricity | 0.239 | 0.148 | 0.243±0.01 | 0.150±0.00 | 0.264±0.01 | 0.170±0.01 |
| Traffic | 0.269 | 0.392 | 0.269±0.01 | 0.394±0.02 | 0.283±0.02 | 0.407±0.02 |
| Exchange | 0.206 | 0.086 | 0.208±0.00 | 0.087±0.00 | 0.210±0.01 | 0.087±0.00 |
| Dataset | TimeGAN | | ADA | | ReAugment | |
| ETTh1 | 0.409±0.02 | 0.390±0.02 | 0.407±0.01 | 0.391±0.01 | **0.396**±0.01 | **0.381**±0.01 |
| ETTh2 | 0.348±0.01 | 0.299±0.01 | 0.347±0.00 | 0.297±0.00 | **0.346**±0.01 | **0.294**±0.01 |
| ETTm1 | 0.392±0.02 | 0.357±0.03 | 0.372±0.01 | 0.336±0.02 | **0.364**±0.01 | **0.328**±0.01 |
| ETTm2 | 0.279±0.01 | 0.190±0.01 | 0.273±0.00 | 0.188±0.00 | **0.263**±0.01 | **0.179**±0.00 |
| Weather | 0.219±0.01 | 0.177±0.01 | 0.222±0.00 | 0.180±0.00 | **0.206**±0.00 | **0.170**±0.00 |
| Electricity | 0.276±0.02 | 0.183±0.01 | 0.241±0.01 | 0.149±0.01 | **0.236**±0.01 | **0.147**±0.01 |
| Traffic | 0.296±0.02 | 0.412±0.03 | 0.268±0.02 | 0.391±0.02 | **0.264**±0.01 | **0.388**±0.01 |
| Exchange | 0.210±0.01 | 0.087±0.00 | 0.209±0.00 | 0.087±0.00 | **0.204**±0.00 | **0.085**±0.00 |

## F    COMPLETE RESULTS FOR PRELIMINARY FINDINGS

Table 15 provides the full cross-dataset results supporting the preliminary finding in Sec 3.2 that sample-wise variance correlates with forecasting difficulty. For each dataset, we report forecasting performance when training on the top-variance and bottom-variance halves of the data, corresponding to the analysis introduced in the main text.

Table 15: **Forecasting results on high-variance vs. low-variance subsets across all datasets.**

| Subset | ETTh1 | | ETTh2 | | ETTm1 | | ETTm2 | | Weather | | Electricity | | Traffic | | Exchange | |
|---|---|---|---|---|---|---|---|---|---|---|---|---|---|---|---|---|
| | MAE | MSE | MAE | MSE | MAE | MSE | MAE | MSE | MAE | MSE | MAE | MSE | MAE | MSE | MAE | MSE |
| Top 50% variance | 0.418 | 0.399 | 0.357 | 0.305 | 0.402 | 0.359 | 0.272 | 0.187 | 0.224 | 0.185 | 0.256 | 0.161 | 0.289 | 0.413 | 0.210 | 0.087 |
| Bottom 50% variance | 0.403 | 0.386 | 0.344 | 0.297 | 0.362 | 0.329 | 0.258 | 0.177 | 0.217 | 0.176 | 0.250 | 0.154 | 0.292 | 0.415 | 0.205 | 0.085 |
| Full training set | 0.405 | 0.387 | 0.350 | 0.301 | 0.377 | 0.341 | 0.272 | 0.186 | 0.219 | 0.178 | 0.239 | 0.148 | 0.269 | 0.392 | 0.206 | 0.086 |

## LLM USAGE STATEMENT

In preparing this manuscript, Large Language Models (LLMs) are employed solely for language refinement, including word choice and grammatical corrections. They are not involved in the conception of research ideas, the design or execution of experiments, the analysis of results, or the development of the manuscript's scientific content.

