# OpenReview forum: "ReAugment: Learning to Augment Few-Shot Time Series with Model Zoo Guidance"
_ICLR.cc/2026/Conference — Submitted to ICLR 2026_

### Official Review · Reviewer_Di2i · 2025-10-30

**Soundness:** 2
**Presentation:** 2
**Contribution:** 3
**Rating:** 6
**Confidence:** 3

**Summary:**

This work proposed ReAugment, a novel data augmentation method driven by reinforcement learning.  ReAugment is different from the existing data augmentation methods and uses the RL mechanism for effective augmentation. ReAugment significantly boosts forecasting performance while maintaining minimal computational overhead by leveraging a learnable policy to transform the overfit-prone samples.

**Strengths:**

1. The viewpoints presented in this work are quite interesting and may bring new insights to the field.

2. The analysis of the existing works in this work is very accurate.

**Weaknesses:**

1. The efficiency issue in the training phase of reinforcement learning should be discussed more. The discussion of Table 6 in the manuscript is overly simplistic. It is suggested that more experiments be conducted to demonstrate the computational cost of ReAugment.

2. The insight of using reinforcement learning for data augmentation is interesting. However, the two objectives of Line 256 are mutually exclusive. Can more discussions and experimental explorations be conducted on these two objectives?

3. As shown in Fig 1, the performance improvement of ReAugment is insignificant in the Exchange. The reasons of this phenomenon need to be analyzed in detail.

**Questions:**

See Weaknesses

---

> ### Author Response · Authors · 2025-11-21
> **Response to reviewer Di2i**
>
> We thank the reviewer for the valuable comments, and we address the concerns in our reply. We remain fully willing to elaborate on any remaining questions.
> > 1. Clarifying the computational overhead of ReAugment by decomposing its cost, explaining why the RL phase is lightweight, and providing explicit cross-method comparisons.
>
>
> We show that ReAugment’s RL component introduces only moderate overhead compared with other augmentation baselines. As shown below, the RL overhead is comparable to learning-based augmentation methods (e.g., TimeGAN and ADA), while hand-crafted augmentations incur almost no training cost but yield weaker gains in our setting. We have added these comparisons to Table 6.
>
> | Dataset | Our Stage A (VMAE)| Our Stage B (RL) | TimeGAN | ADA |
> |---------|---------|---------|---------|-----|
> | ETTh1   | 1min    | 2min    | 4min      | 1min  |
> | Elec.   | 24min   | 31min   | 1h12min      | 18min  |
> | Traffic | 1h 22min| 1h 53min| 3h47min      | 58min  |
>
> Notably, ReAugment’s extra cost mainly comes from the RL phase that adjusts the latent prior. We claim that this phase is lightweight because the VMAE decoder and all forecasting models are frozen; RL only performs latent sampling, a single decoding step, and forward evaluations of the model zoo. No backpropagation occurs through the forecasting models, and the policy is optimized entirely in the low-dimensional latent space.
>
> > 2. Explaining why the two RL objectives are not contradictory and how their trade-off reflects the core principle of effective data augmentation.
>
> Although the reward encourages both higher model-zoo variance and proximity to the original sample, these objectives are not conflicting; they jointly capture the local instability structure around overfit-prone regions.
> - Model-zoo variance reflects forecasting sensitivity rather than Euclidean distance, so small, realistic perturbations near difficult anchors can substantially increase predictive disagreement without drifting off-distribution.
> - The fidelity term ensures that augmentations remain close to the data manifold, guiding RL toward controlled, high-impact modifications rather than large, noisy deviations.
>
> The core significance of data augmentation is the trade-off between the two goals.
>
>
> > 3. Explaining why ReAugment yields smaller gains on Exchange due to its low overfitting risk and limited performance margin.
>
> Unlike other datasets, baseline forecasting models already generalize well on Exchange even with limited early data. This is likely because Exchange is more stationary and exhibits weaker overfitting. As a result, the “recoverable” performance margin is inherently small. This is also reflected in our F-metrics: although the absolute improvement in MSE/MAE is small, the improvement in F-metrics is substantial. Consequently, even when ReAugment closes most of the available gap, the absolute improvement naturally appears less pronounced.

---

### Official Review · Reviewer_9veJ · 2025-10-31

**Soundness:** 2
**Presentation:** 2
**Contribution:** 3
**Rating:** 4
**Confidence:** 4

**Summary:**

This paper presents a RL based method to data augmentation for time-series forecasting tasks. The method includes the use of cross-validation errors for identifying potentially data samples prone to overfitting, use this samples to anchor VAE-based augmentation, and use this VAE as an actor in RL to derive the augmentation policy guided by the goal to increase diversity while remaining close to the original samples. Experiments were performed on five public time series datasets for few-shot as well as standard training settings, in comparison to several alternative data augmentation methods.

**Strengths:**

- The presented idea is overall novel and interesting, especially to learn augmentation policy as a RL problem.

- The use of cross validation (under the name of model zoo) to identify overfitting-prone samples has merits.

- The analysis showing the impact of the percentage of data augmented (Fig 4a) is insightful, especially the observation that augmenting all data is less effective.

**Weaknesses:**

- The design of the RL formulation is not well justified in terms of the underlying MDP and the justification for the choices of the state and reward functions. It is not clear what are the states, or why the use of masked time series data and timestamps are sufficient as state variables. The definition of Var in Equation 3 is also very unclear and, more importantly, it is curious that the reward function does not seem to care about the forecasting model performance on the augmented data (but only the diversity and fidelity of the augmented samples). It seems that it is possible that such an augmentation policy could lead to the augmentation samples that will deteriorate the forecasting function, as the augmentation is not guided by this important goal.

- The use of the VAE as the actor is also not well rationalized — why is the prior network chose as the policy network to be optimized in RL (instead of any other components, such as the decoder that is actually generating the augmented samples).


- Statistics from multiple runs should be included for all baselines and experiments, not just for the presented method, especially in datasets where the margins of improvements are small.

**Questions:**

- The design choices for the MDP underlying the RL formulation need to be clearly motivated, described, and experimentally verified (e.g., the choice of which part of the VAE is considered as the policy network, what states to use, etc).

- Adding statistics from multiple runs for statistical significance of the presented method.

---

> ### Author Response · Authors · 2025-11-21
> **Response to reviewer 9veJ (Part 1)**
>
> We truly appreciate the reviewer’s insightful comments, and we respond to each point in the following. We are happy to offer additional details if needed.
>
> > 1. Justifying the RL formulation—its contextual-bandit MDP, state design, and variance-based reward—and explaining how the reward implicitly encodes forecasting performance while preventing harmful augmentations.
>
> Below we provide a concise clarification of the RL formulation and design choices.
>
> (1) *Underlying MDP (actually a contextual bandit)*
>
> The RL part of ReAugment is **one-step**, i.e., a contextual bandit:
>
> |Component|Definition|
> |-|-|
> |**State**|Masked anchor sequence $m_{1:L}$ + timestamps $t_{1:L}$|
> |**Action**|Sample latent code $\tilde{z} \sim p(\tilde{z} \mid m_{1:L}, t_{1:L})$ from the VMAE **prior**|
> |**Transition**|Generate $\hat{s}_{1:L} = \text{Dec}([u,\tilde{z}])$; episode ends immediately|
> |**Reward**|Sigmoid-transformed score based on model-zoo variance and fidelity (Eq. (4))|
> |**Horizon**|$T=1$, $\gamma=0$|
>
> We have explicitly described this in the revision.
>
> (2) *It is not clear what are the states, or why the use of masked time series data and timestamps are sufficient as state variables.*
>
> The state input $(m_{1:L}, t_{1:L})$ is **the same conditioning information** used in the VMAE reconstruction in Stage A. It captures:
>
> - local temporal patterns (via the masked sequence $m_{1:L}$)
> - absolute temporal context (via timestamps $t_{1:L}$).
>
> Because our RL is one-step, augmentations are **conditionally independent across anchors** given this local context. Adding additional history or global information would not change the structure of the policy’s decision (it still chooses a latent $\tilde{z}$ once per anchor), while substantially increasing state dimensionality and noise. Empirically, richer context did not improve performance, whereas the current choice yields stable and effective augmentations.
>
> (3) *What Var in Eq. (3) actually measures*
>
> We apologize for the confusion caused by the original notation. Our intention is to compute the **variance of forecasting errors across the model zoo** for each augmented sample. Let
>
> - $\hat{y}\_k(\hat{s}\_{1:L})$ be the **forecasting output** produced by the $k$-th model $\omega_k$ when fed the augmented sequence $\hat{s}_{1:L}$, over the same prediction horizon used in backtesting
> - $y(\hat{s}_{1:L})$ be the corresponding ground-truth target for that horizon.
>
> Then,
> $e\_k = \big\|\hat{y}\_k(\hat{s}\_{1:L}) - y(\hat{s}\_{1:L})\big\|\_2^2, \qquad
> \bar{e} = \frac{1}{K-1}\sum\_{k=1}^{K-1} e\_k,$
>
> $\mathrm{Var}(\hat{s}\_{1:L}; M)
> = \frac{1}{K-1}\sum\_{k=1}^{K-1}(e\_k - \bar{e})^2.$
>
> Thus, $\mathrm{Var}(\hat{s}_{1:L}; M)$ measures **how differently the forecasting model zoo react to the same augmented input**, identifying regions where prediction behaviors diverge—typically overfit-prone or unstable areas. We will update Eq. (3) and the surrounding text to use this clearer notation and explicitly refer to the forecasting outputs $\hat{y}_k$.
>
> We have corrected Eq. (3) in the revision.
>
> (4) *Clarifying how the reward implicitly encodes forecasting performance through model-zoo error variance while preventing degenerate or harmful augmentations.*
>
> We appreciate the reviewer’s concern. The goal of the reward design in ReAugment is *not* to ignore forecasting performance, but to incorporate it in a way that avoids degenerate behavior. Importantly, the reward **does** depend on forecasting performance—just not through a single forecasting model’s loss, but through the *model-zoo prediction errors*, which offer a more reliable signal in few-shot and temporally shifted settings.
>
> Concretely, the variance term $\mathrm{Var}(\hat{s}_{1:L};M)$ is computed from the forecasting **prediction errors** of a diverse set of models on the same augmented sample. A sample that consistently leads to poor or unstable forecasting behavior across the zoo will yield *high error variance*, and thus a *high reward* **only if** it remains close enough to a realistic sequence (controlled by the fidelity term in the denominator). Conversely, samples that are unrealistic or harmful tend to produce *large errors for all models uniformly*, resulting in **low variance**, and thus **low reward**. In this sense, the reward *implicitly but robustly* filters out harmful augmentations by examining how models behave collectively, rather than relying on a single forecasting model’s loss.
>
> We also note that directly rewarding low forecasting loss introduces an opposite problem: it encourages the policy to generate extremely simple or easily predictable samples, which do not enrich the training distribution and empirically lead to collapsed or uninformative augmentations. By contrast, using the *spread* of model errors (instead of the absolute error) identifies regions where models disagree—i.e., areas that are informative but not adversarial—and the fidelity constraint prevents the policy from drifting into unrealistic regimes.

---

> ### Author Response · Authors · 2025-11-21
> **Response to reviewer 9veJ (Part 2)**
>
> > 2. The use of the VAE as the actor is also not well rationalized — why is the prior network chose as the policy network to be optimized in RL (instead of any other components, such as the decoder that is actually generating the augmented samples).
>
> This is the same input used in VMAE reconstruction (Stage A) and is empirically sufficient. Because the RL is one-step, adding more history does not change the decision structure and only adds noise.
>
> **Reason 1 — Training Stability.**
> - **Prior as an Actor:** The prior network determines where to sample in latent space conditioned on the anchor input. This aligns exactly with the function of a policy in reinforcement learning — selecting an action (latent perturbation) given a state (the anchor). Moreover, the prior outputs a compact and smooth action space (low-dimensional, Gaussian), making it significantly easier and more stable to optimize under one-step rewards compared to high-dimensional decoder outputs.
> - **Decoder as a fixed Generator:** The decoder defines semantic structure of latent space learned in Stage A. Updating it with noisy RL gradients would break this mapping.
>
> **Reason 2 — Empirical Evidence (New Experiments Added)**
>
> In Table 5 of the revised manuscript, we further conducted controlled experiments comparing three choices for the policy network::
> - The decoder,
> - The entire VMAE (encoder + prior + decoder),
> - Our default design: the prior alone.
>
> As shown below, the prior-based policy consistently achieves the strongest performance across all datasets, while the decoder–based and full-VMAE variants yield lower accuracy under the same setting. These results indicate that using the prior as the policy network is the most effective design choice among the tested alternatives, and therefore we adopt this configuration in all main experiments.
>
> | Dataset        | Metric | Decoder | Full VMAE | Prior (ours) |
> |----------------|--------|---------|-----------|---------------|
> | **ETTh1**      | MAE    | 0.429   | 0.424     | **0.422**     |
> |                | MSE    | 0.407   | 0.404     | **0.403**     |
> | **ETTh2**      | MAE    | 0.350   | 0.341     | **0.339**     |
> |                | MSE    | 0.315   | 0.304     | **0.302**     |
> | **ETTm1**      | MAE    | 0.421   | 0.416     | **0.410**     |
> |                | MSE    | 0.444   | 0.441     | **0.436**     |
> | **ETTm2**      | MAE    | 0.277   | 0.276     | **0.275**     |
> |                | MSE    | 0.198   | **0.196** | **0.196**     |
> | **Weather**    | MAE    | 0.231   | 0.230     | **0.229**     |
> |                | MSE    | 0.187   | 0.186     | **0.185**     |
> | **Electricity**| MAE    | **0.254** | 0.255   | **0.254**     |
> |                | MSE    | 0.165   | 0.166     | **0.165**     |
> | **Traffic**    | MAE    | 0.295   | **0.293** | **0.293**     |
> |                | MSE    | 0.432   | 0.430     | **0.429**     |
>
>
> > 3. Ensuring fair comparison by reporting standard deviations for all methods.
>
> In Appendix E in our original submission, we have provided the mean and standard deviation for all baseline methods evaluated over three random seeds. Additionally, in the revision, we include the corresponding standard deviations for the newly introduced metrics to ensure full consistency and transparency.

---

### Official Review · Reviewer_CATJ · 2025-10-31

**Soundness:** 3
**Presentation:** 3
**Contribution:** 3
**Rating:** 4
**Confidence:** 2

**Summary:**

This paper addresses the limited availability of high-quality training data. An RL-based method is proposed that 1) overfit-prone data are identified as augmentation anchors, 2) a tailored VMAE for sequential data produces sample-aware augmentations, and 3) an RL approach is used to train the augmentation network guided by a reward function derived from a forecasting model zoo. Experiments have been conducted to prove the effectiveness of the proposed method compared with various base models.

**Strengths:**

- The problem of limited high-quality training data is critical, and the idea of identifying the overfit-prone samples is important.
- The description of the methodology is well-structured.
- The experiments showed the improvement of the proposed method.

**Weaknesses:**

- More details of the methodology need to be disclosed.
- The experimental results and the proposed new metrics need to be justified.

**Questions:**

- Section 3.1: What’s the motivation for two subsets of the training set? How would the choices of the subsets affect the performance?
- Section 3.2: The authors state that the overfit-prone samples are more likely to negatively affect the training quality of forecasting models. Is this statement universal across other datasets?
- The improvement of the proposed method in Table 1 and 2 is not significant. Could the authors provide visualizations to justify the metric numbers? Also, it would be more consistent to have the variance of all metrics available in tables.
- Could the authors elaborate on the new metrics proposed in Section 5.2? It is not clear what is expected for this metric and the meaning of negative values in Table 3.

---

> ### Author Response · Authors · 2025-11-21
> **Response to reviewer CATJ (Part 1)**
>
> We are grateful for the reviewer’s constructive feedback, and we provide our responses below. We would be glad to offer additional details if needed.
> > Weaknesses: More details of the methodology need to be disclosed; The experimental results and the proposed new metrics need to be justified.
>
> Please see our following responses.
>
> > 1. Section 3.1: What’s the motivation for two subsets of the training set? How would the choices of the subsets affect the performance?
>
> We have provided additional clarifications in Section 3 in the revised manuscript.
>
> The two subsets (top 50% high-variance and bottom 50% low-variance) are introduced for a single purpose: **to validate that the proposed '*model-zoo variance*' is a meaningful indicator for detecting the '*overfit-prone regions*' in the training set**, which directly motivates our anchor-selection strategy (anchors refer to the input data of the generative augmentation model).
>
> We here provide more detailed explainations:
>
> - Motivation: We compute model-zoo variance as the disagreement of K fold-trained forecasting models on each training window. High variance indicates regions where models are unstable and prone to overfitting; low variance indicates stable, well-supported regions. This behavior is empirically confirmed in Fig. 2.
> - Why two subsets: Splitting into the top/bottom 50% allows a clean diagnostic --- Training only on low-variance samples yields much better generalization than training only on high-variance ones. This demonstrates that high-variance samples are precisely the difficult areas where augmentation should concentrate. The 50% split is a simple midpoint used only for analysis. It is sufficient to reveal the variance-based difficulty structure and provide a reasonable anchor pool.
> - Effect on performance: In the full pipeline, we therefore augment only the high-variance subset. Fig. 4 shows that targeted augmentation on this subset outperforms augmenting the entire dataset—because high-variance samples benefit from synthetic support, while low-variance samples are already stable.
>
>
> > 2. Section 3.2: The authors state that the overfit-prone samples are more likely to negatively affect the training quality of forecasting models. Is this statement universal across other datasets?
>
> Yes, the effect described in Section 3.2 is consistent across datasets. In the revised manuscript, we provide empirical evidence across all eight benchmarks in Table 15 to support this claim. We summarize the results below:
>
> ||Top 50\% variance data|Bottom 50\% variance data| complete training set|
> |-|-|-|-|
> |ETTh1|0.418|0.403|0.405|
> |ETTh2|0.357|0.344|0.350|
> |ETTm1|0.402|0.362|0.377|
> |ETTm2|0.272|0.258|0.272|
> |Weather|0.224|0.217|0.219|
> |Traffic|0.289|0.292|0.269|
> |Electricity|0.256|0.250|0.239|
> |Exchange|0.210|0.205|0.206|
>
> Across all benchmarks, models trained exclusively on the top-variance half consistently underperform those trained on the bottom-variance half, indicating that high-variance samples increase instability and hinder generalization. Datasets with stronger non-stationarity (e.g., ETTh1, ETTm1, Electricity) exhibit more pronounced degradation on high-variance subsets, while more stationary datasets (e.g., Exchange) show smaller but consistent difference, confirming the universality of the observation in Section 3.2.

---

> ### Author Response · Authors · 2025-11-21
> **Response to reviewer CATJ (Part 2)**
>
> > 3. The improvement of the proposed method in Table 1 and 2 is not significant. Could the authors provide visualizations to justify the metric numbers? Also, it would be more consistent to have the variance of all metrics available in tables.
>
> (1) Visualizations corresponding to Tables 1.
>
> To make the performance differences easier to interpret, we have added bar-chart visualizations corresponding to Tables 1  in the revised manuscript (Figure 4), where each dataset  is plotted for all compared methods. These visualizations highlight the relative ranking more clearly than raw numbers and make the improvements visually evident.
>
> For completeness, the detailed numerical results remain in Appendix E, and we additionally report the variance of all metrics across repeated runs, as suggested.
>
> (2) Why improvements appear small but are meaningful.
>
> Few-shot multivariate forecasting under severe temporal shift is a highly constrained regime, where base models rapidly reach a performance plateau due to the structural gap between early training data and much later test periods. In such settings, any additional improvement tends to be moderate in absolute value.
> More importantly:
>
> - ReAugment produces consistent gains across 8 datasets, 3 forecasting architectures, and both MAE/MSE metrics.
> - These gains hold in both few-shot and standard settings (Tables 1–4).
> - Table 3 shows that, when normalized by the “full-data gap,” ReAugment often recovers a substantial portion of the missing performance—demonstrating that the method is impactful relative to the inherent difficulty of the task.
>
> Thus, although the numerical deltas appear small, their consistency and universality indicate that ReAugment meaningfully improves generalization in a regime where further gains are inherently hard to obtain.
>
> (3) Variance results.
>
> In Appendix E, we provide the mean and standard deviation for all baseline methods evaluated over three random seeds. In the revised version, we also include the corresponding standard deviations for the newly introduced metrics.
>
> > 4. Could the authors elaborate on the new metrics proposed in Section 5.2? It is not clear what is expected for this metric and the meaning of negative values in Table 3.
>
> The metrics introduced in Section 5.2 are designed to quantify how much of the few-shot performance gap an augmentation method recovers relative to a full-data model. Specifically, each F-metric computes the fraction of the recoverable improvement achieved:
> - A value of **100\%** means the method closes the entire gap to full-data performance,
> - **>100\%** indicates surpassing full-data performance,
> - **0** means no improvement,
> - **Negative values** indicate that the augmentation is **harmful**.
>
> Negative entries in Table 3 therefore reflect augmentation methods (e.g., noise, jitter, or unconstrained generative approaches) that distort temporal structure and degrade generalization. The F-metrics thus provide a normalized and interpretable scale: even a small reduction in MAE/MSE can correspond to recovering a substantial portion of the full-data performance gap—information that raw errors alone do not reveal.

---

### Official Review · Reviewer_S6M8 · 2025-11-03

**Soundness:** 3
**Presentation:** 3
**Contribution:** 2
**Rating:** 4
**Confidence:** 3

**Summary:**

This paper deals with the problem of time-series forecasting and proposes a method for augmenting training time-series examples. More specifically, the proposed method first trains a variational masked autoencoder (VMAE) with the original training examples and then chooses promising training examples that can be a source of augmentation. The proposed example selection introduces a training scheme inspired by a policy gradient method, and its reward is defined by variance of predictions obtained from a group of time-series estimators. In this framework, the distribution parameter predictor of the VMAE can be regarded as a "policy network" in the context of reinforcement learning.

**Strengths:**

S1. The problem dealt with in this paper is significant. Optimizing data augmentation schemes is one of the significant components for obtaining sufficient prediction performances in low-resource regimes.

S2. The strategy of the proposed method is reasonable and technically sound. Reinforcement learning is one of the standard approaches for automatically optimizing data augmentation schemes. The proposed method incorporates the idea of REINFORCE, one of the standard policy gradient methods, into this optimization. Also, focusing on prediction variances as a measure for "goodness" in data augmentation is reasonable.

S3. This paper is well-written and easy to follow.

**Weaknesses:**

W1. Novelty of the proposed method against several previous methods should be justified literally and hopefully its effectiveness should have been demonstrated by experimental comparisons. One of the key ideas of the proposed method is to choose promising training examples useful for data augmentation, which have already been explored in the following papers:

  - [Yang+ ICCV2025 https://arxiv.org/abs/2506.21037] (Note: This paper was officially published AFTER the ICLR2026 deadline, which indicates that this paper should not be a source of loss of novelty.)
  - [Huang+ IEEE TGRS 2024 https://ieeexplore.ieee.org/document/10714383]

Although I understand that details of the proposed method is different from those of the above previous methods, novelty of the proposed method and its effectiveness should have been justified.

W2. I could not understand the reason why only the proposed method has standard deviation in the experimental results. Every augmenter has randomness, which indicates that experimental results can be variable according to the randomness. Presenting the deviation is significant for understanding statistical significance for the presented experimental results.

**Questions:**

None.

---

> ### Author Response · Authors · 2025-11-21
> **Response to reviewer S6M8**
>
> We sincerely appreciate the reviewer’s thoughtful comments, and we address them in our response. Please let us know if any further clarification would be helpful.
> > 1. Justifying the novelty of our RL-based generative augmentation method beyond prior sample-selection approaches.
> >
> We thank the reviewer for pointing us to these related work. We have incorporated concise clarifications and comparisons in the revised Related Work section.
>
> Although ADAAUG+ and RL-Selector also use RL to prioritize or select samples, their tasks, goals, and outputs differ fundamentally from ours (as shown below). While all three methods involve '*sample prioritization*', ours uniquely focuses on anchor-guided generative augmentation for time series forecasting, rather than operation selection or sample pruning.
> - Different tasks and goals: ADAAUG+ searches for augmentation operations in remote-sensing change detection, and RL-Selector identifies redundant samples for pruning in image classification. In contrast, ReAugment is a generative framework specifically for few-shot time-series forecasting, aiming not to select or prune samples but to synthesize new task-aware sequences that enrich the forecasting distribution.
> - Different definitions and uses of "promising samples": Previous works use heuristic-based indicators for pruning or reweighting. We introduce model-zoo prediction variance as a principled indicator of overfit-prone regions and use these samples as anchors for generation—not for removal. Figure 2 and 4 confirm that augmenting only high-variance anchors is substantially more effective than uniform augmentation.
> - Different RL formulation and action space: Both ADAAUG+ and RL-Selector use RL in discrete or binary spaces (operation selection or keep/discard). In contrast, ReAugment optimizes directly in a continuous VMAE latent space, and each action corresponds to a complete synthetic time series. The reward integrates (i) model-zoo disagreement and (ii) fidelity constraints, enabling task-aware generative augmentation—a capability fundamentally absent in previous works.
>
> | Aspect       | ADAAUG+                         | RL-Selector             | ReAugment (ours)                     |
> |--------------|---------------------------------|-------------------------|--------------------------------------|
> | Task         | Remote-sensing change detection | Image classification    | Few-shot time-series forecasting     |
> | Goal         | Choose augmentation operations         | Prune redundant samples | Generate task-aware time series     |
> | Action Space | Discrete operations                    | Binary keep/discard     | Continuous VMAE latent codes         |
> | Output       | Operation-selection policy             | Smaller dataset         | Expanded dataset via generation |
>
> The effectiveness of these design choices is supported across models and datasets (Tables 1–4, Fig. 4–5).
>
>
> > 2. Ensuring fair comparison by reporting standard deviations for all methods.
>
> In Appendix E in the original manuscript, we have provided the mean and standard deviation for all baseline methods evaluated over three random seeds. In the revised paper, we further include the corresponding standard deviations for the newly introduced metrics to ensure full consistency and transparency.

---

### Author Response · Authors · 2025-11-21
**Revision uploaded**

We have updated the manuscript to incorporate the following revisions:
1. The related work section now provides strengthened comparisons to prior sample-selection approaches (Sec. 2).
2.  We add an explicit justification showing that model-zoo variance empirically separates stable and unstable regions, strengthening the rationale for the anchor-selection criterion (Sec. 3.1).
3. The preliminary findings section now offers a clearer causal interpretation linking low-variance subsets to improved model stability and generalization (Sec. 3.2).
4. We expand the justification for the RL formulation (Sec. 4.3).
5. We correct the variance-reward formula in the REINFORCE-based augmentation section to use only the second half of the sequence $(L/2 : L)$ as the prediction target, fixing the mathematical definition (Sec. 4.3).
6. We convert the original Table 1 into a bar chart to better visualize the error bars (Fig. 4), while keeping the full numerical results in Appendix E (Sec. 5.2 & Appendix E).
7. The titles of Tables 2–3 now include additional notes indicating that Appendix E contains the full experimental results with means and standard deviations (Sec. 5.2 & Appendix E).
8. We add an explanation to clarify how to interpret the different value ranges of the new metric (Sec. 5.2).
9. We include additional experiments to validate why only the prior network is used as the policy network (Sec. 5.5).
10. We clarify the computational overhead of ReAugment and provide explicit cross-method comparisons (Sec. 5.5).
11. We add a table reporting experimental results for the new metrics, including means and standard deviations (Appendix E).
12. We add the complete experimental results of the preliminary findings across all datasets (Appendix F).

---

### Author Response · Authors · 2025-12-03

We thank the Area Chair for taking over the evaluation of our submission and for considering our revision alongside the original reviewer comments. To streamline the assessment, we concisely summarize (i) the central idea and technical novelty of ReAugment, (ii) the three recurring methodological concerns raised across reviews, and (iii) the major clarifications, experiments, and revisions incorporated in the updated manuscript.

## 1. Summarization of Technical Novelty

ReAugment introduces a variance-guided, RL-driven generative augmentation framework for few-shot time-series forecasting.
Its core components are:
- Variance-guided anchor selection: Using a forecasting model zoo, we compute prediction-error variance to detect overfit-prone samples.
- Generative policy in latent space: A VMAE backbone provides a continuous, structured latent space. The prior network is optimized as the policy in a one-step contextual bandit to generate realistic yet variance-increasing augmentations.
- Closed-loop augmentation–forecasting alignment: Model-zoo predictions supply task-aware reward signals that guide augmentation toward regions where forecasting models benefit most.

This pipeline aims to produce task-aligned synthetic sequences that enhance stability and generalization under data scarcity without altering the forecasting architecture.

Across all reviews, the major questions converge into three common themes, which directly determine the validity and impact of the method:
> 1. Reliability of Model-Zoo Variance as the Anchor Criterion

Reviewers ask whether model-zoo variance is a meaningful and universal signal for identifying overfit-prone regions. To address this, we provide complete diagnostics across all eight datasets (Appendix F) showing that the bottom-variance half always yields stronger generalization than the top-variance half, confirming that variance consistently separates stable from unstable regions. This validates our anchor-selection strategy, which is foundational to the entire augmentation pipeline.

> 2.  RL Formulation, State/Action Definition, and Reward Design
>
Reviewers request clearer justification of the contextual-bandit formulation, the state/action definition, and the variance-based reward. We therefore supply an explicit one-step MDP description (state = masked anchor + timestamps; action = latent code), correct the variance-reward equation, and clarify that the reward is computed from prediction-error variance across the model zoo, which implicitly encodes forecasting performance while the fidelity term prevents harmful off-manifold samples. This explains why RL is suitable and how it avoids the degeneration issues faced by loss-based backpropagation through the zoo.

> 3. Policy Design: Why Only the Prior Network is Optimized
>

We add policy ablations (Table 5) showing that the prior-only policy consistently achieves the best performance, while decoder-policy or full-model-policy variants are less stable. We explain that the prior provides a smooth, low-dimensional, anchor-conditioned action space, aligning precisely with the function of a policy and avoiding disruption of the decoder’s learned semantic structure.

## 2. Key Improvements and New Experiments in Revision
To make the results clearer, more interpretable, and more statistically sound, we additionally:
- introduce bar-chart visualizations for main results (Fig. 4);
- provide means and standard deviations for new metrics in Appendix E;
- provide more detailed computational cost evaluation computational overhead, showing RL adds only moderate cost relative to other generative augmentations (Table 6);
- ensure all results referenced in the main paper are linked to complete tables in the Appendix, where every metric across all baselines is reported with full statistical summaries.

## 3. Final Remarks
We thank the Area Chair for taking the time to evaluate our submission. This revision presents a concise clarification of the core components of ReAugment, including the variance-based anchor selection, the reinforcement learning formulation, and the rationale for optimizing only the prior network as the policy. We provide additional diagnostics, ablations, and experimental results to address the reviewers’ main concerns and to give a clear, verifiable account of the method. We hope that the organized structure of the revision supports an efficient assessment of the technical validity and empirical contribution of the work.

---

### Meta-Review · Area_Chair_Ri1E · 2025-12-15

**Summary:**

This paper proposes ReAugment, a reinforcement-learning-based framework for data augmentation in few-shot time series forecasting. The method identifies overfit-prone samples using prediction variance across a model zoo and learns a generative augmentation policy guided by downstream forecasting performance.

The reviewers generally agree that the problem setting is relevant and that the paper is well written and technically competent. The idea of leveraging model disagreement to guide data augmentation is intuitive, and the authors demonstrate substantial effort in responding to reviewer feedback through clarifications, additional experiments, and improved presentation.

However, despite these strengths, there remain several fundamental concerns that prevent the paper from meeting the acceptance bar of this venue. S6M8, CATJ, and 9veJ independently raised concerns about novelty, the necessity of RL, and limited empirical impact. While the paper is competently executed, reviewer consensus indicates that it does not meet the novelty and significance bar for acceptance.

In summary, this paper addresses an important problem and demonstrates solid execution and responsiveness to reviewer feedback. However, the contribution is incremental, the role of reinforcement learning is not fully compelling, and the empirical gains do not clearly justify acceptance at this venue. So I give my recommendation as reject.

**Reviewer Concerns:**

In my view, the rebuttal successfully improves clarity and empirical support, but core concerns regarding novelty, the necessity of RL, and overall impact remain unresolved, supporting a Reject decision.

**Reviewer Scores:**

Considering that core concerns regarding novelty, the necessity of RL, and overall impact remain unresolved, I don't think reviews will shift their rate to acceptance, leaving the final consensus unchanged.

---

### Decision · Program_Chairs · 2026-01-26

Reject